# Effects of Paper Mulberry Silage on the Milk Production, Apparent Digestibility, Antioxidant Capacity, and Fecal Bacteria Composition in Holstein Dairy Cows

**DOI:** 10.3390/ani10071152

**Published:** 2020-07-07

**Authors:** Yangyi Hao, Shuai Huang, Jingfang Si, Jun Zhang, Naren Gaowa, Xiaoge Sun, Jiaying Lv, Gaokun Liu, Yaqin He, Wei Wang, Yajing Wang, Shengli Li

**Affiliations:** State Key Laboratory of Animal Nutrition, Beijing Engineering Technology Research Center of Raw Milk Quality and Safety Control, College of Animal Science and Technology, China Agriculture University, Beijing 100193, China; haoyangyi0928@163.com (Y.H.); huangshuai510@126.com (S.H.); sijingfang@foxmail.com (J.S.); june_zh16@cau.edu.cn (J.Z.); narengaowa@cau.edu.cn (N.G.); xiaogesun@163.com (X.S.); b20183040337@cau.edu.cn (J.L.); liugk339@163.com (G.L.); heyaqin1184726515@126.com (Y.H.); wei.wang@cau.edu.cn (W.W.); yajingwang@cau.edu.cn (Y.W.)

**Keywords:** dairy cow, paper mulberry, production, antioxidant capacity, fecal bacteria

## Abstract

**Simple Summary:**

Paper mulberry (*Broussonetia papyrifera;* PM) is a type of roughage rich in bioactive substances, such as phenolics and flavonoids, which are beneficial for animal health. This study evaluated the apparent digestibility of PM silage in Holstein dairy cows and its effect on the milk production, antioxidant capacity, and fecal bacteria composition of the animals. The results showed that the PM silage had no significant influence on the milk yield, apparent digestibility, and fecal bacteria composition of dairy cows. However, diets with PM silage can enhance the antioxidant and immune capacity of dairy cows, mainly due to the bioactive substance in PM. Today, faced with a shortage of feedstuff resources in ruminants, PM can be a useful feed resource for ruminants. Simultaneously, with the ban on antibiotics, PM may become an important functional feed for protecting animal health.

**Abstract:**

Paper mulberry (*Broussonetia papyrifera*; PM) is an excellent and extensive type of roughage in Asia. This study aimed to evaluate the effects of PM silage on the milk production, apparent digestibility, antioxidant capacity, and fecal bacteria composition in Holstein dairy cows. Forty-five lactating Holstein dairy cows with a similar milk yield and parity were selected and randomly assigned to three groups. The control group was fed a non-PM silage diet, and the PM-treated groups were fed 4.5 and 9.0% PM silage supplementary diets for 28 days. Then, treatment groups were fed diets containing 13.5 and 18.0% PM silage for the next 28 days, respectively. PM silage increased the milk urea nitrogen and decreased the somatic cell count (*p* < 0.05), but did not affect the dry matter intake, milk yield, apparent digestibility, and energy balance of dairy cows. PM silage can enhance the blood total antioxidant capacity, superoxide dismutase, and immune globulin content (*p* < 0.05). The PM silage significantly decreased the relative abundance of the genera *Ruminococcaceae UCG-013* and *Tyzzerella-4* (*p* < 0.05). In conclusion, PM silage enhanced the antioxidant capacity and immunity of dairy cows, but did not influence the milk yield, dry matter digestibility, and fecal bacteria composition.

## 1. Introduction

Paper mulberry (*Broussonetia papyrifera*; PM) is a dioecious tree native to mainland South-Asia and East Asia [1]. Twenty years ago, PM was planted to prevent soil erosion in China. In recent years, it has been recognized that PM grows rapidly and has a high protein content, which makes it useful as a roughage resource for ruminant animals. Currently, about 300,000 hectares of PM plants are under cultivation in China, and most of them are planted as forage crops intended to be processed into silage. Usually, the leaves of PM are fed to pigs and chickens, while the whole plant is processed into silage which is fed to ruminants. The PM can be harvest three to five times per year in China. Furthermore, the process techniques are mature for PM silage. It is documented that mulberry is a high protein form of roughage and can be processed into high-quality silage with additives [2]. After wilting, the mulberry can have superior effects on the non-structural carbohydrate and degrade the structural carbohydrate [3].

In addition to the good nutritional quality and qualified silage fermentation, mulberry also has many bioactive chemical compounds. It is rich in polyphenols such as flavonols, anthocyanins, benzoic acid, and hydroxycinnamic acid [4]. Mulberry extract supplements ameliorate the inflammation-related hematological parameters in carrageenan-induced arthritic rats [5]. The anthocyanins separated from mulberry fruits can scavenge free radicals and inhibit low-density lipid oxidation [6]. Mulberry leaf silage is a qualified feed with a robust antioxidant capacity [2]. High-yielding cows, which are in the peak lactation period [7], can experience a higher metabolic intensity, with a vulnerable antioxidant and immune capacity [8]. Antioxidant ingestion can enhance the health of cattle. Diets with added shrubs or herbs, which are rich in bioactive substances, can improve the animal’s antioxidant capacity [9]. A higher antioxidant capacity can neutralize oxygen free radicals and remove hydrogen peroxide and superoxide ions [6,10]. With the ban on antibiotics, animals need more protection to keep them healthy. Therefore, we hypothesized that the PM silage could improve the antioxidant capacity of high-yield cows.

The gastrointestinal bacteria composition is relevant to the metabolism status [11] and production performance of host animals [12]. A previous survey suggested that bovine fecal bacteria community structures can dramatically differ at the phylum and family levels, depending on the animal feed [13]. Diet appeared to have a significant effect on the composition of fecal bacteria composition, particularly when comparing forage and grain diets [14]. However, there is no report on the impacts of diets supplemented with PM silage on the fecal bacteria composition in Holstein dairy cows. It has been reported that polyphenols can regulate the intestinal bacteria composition and inhibit the growth of pathogenic bacteria [15]. PM is rich in polyphenols [4]. Therefore, we also hypothesized that diets supplemented with PM silage could manipulate the fecal bacteria composition of Holstein dairy cows.

Therefore, the objective of our study was to evaluate the apparent digestibility of PM silage in Holstein dairy cows and its effect on the milk production, antioxidant capacity, and fecal bacteria composition of the animals.

## 2. Materials and Methods

### 2.1. Treatments, Experimental Design, and Cow Management

The feeding experiment was conducted at Aomei Dairy Farm in Xinxiang City, Henan province, China, using dairy cows with an average 305-day milk yield per cow, milk fat content, milk protein content, and somatic cell count (SCC) of 9000 kg, 3.9%, 3.4%, and 20,000/mL, respectively. All experimental procedures were approved by the Ethical Committee of the College of Animal Science and Technology of China Agriculture University (Protocol number: 2013-5-LZ). The 2nd parity is the most common stage of lactating dairy cows in China. Therefore, forty-five healthy Holstein dairy cows (days in milk: 90 ± 15; daily milk yield: 36.0 ± 1.2 kg/day; parity: 2.0 ± 0.4; body weight: 650 ± 37 kg) were randomly assigned to three groups and housed in three separate free-stall barns. To evaluate the effects of feeding different levels (4.5, 9.0, 13.5, and 18.0%) of PM silage on dairy cows, the experiment was divided into two periods. The length of the adaption period was 7 days, and the formal experimental period was 21 days for both periods. In the first period (Period I), the control and treatment groups were fed total mixed ration (TMR) with 0 (CON), 4.5 (PM1), or 9.0% (PM2) PM silage, respectively. In the second period (Period II), the amount of PM silage was increased to 13.5 and 18.0% in the PM1 and PM2 groups, respectively. The chemical compositions of PM silage and other fodders were analyzed once every 14 days. The average chemical composition of PM silage is presented in Appendix A, while the ingredients and nutrient levels of the diets are presented in Table 1. All cows had access to feed and water ad libitum. Cows were fed two times/day (06:00 and 18:00) and milked two times/day (05:30 and 17:00). Milk yield data was recorded manually every day.

### 2.2. Sample Collection and Analysis

#### 2.2.1. Collection and Analysis of Milk Samples

About 50 mL of milk sample was collected on 14th and 28th day for each period. The milk protein, fat, lactose, milk urea nitrogen (MUN), and SCC were analyzed by an automated near-infrared milk analyzer (CombiFoss FT+; Foss Electric, Hillerød, Denmark) at the Henan DHI Testing Center (Zhengzhou, China).

#### 2.2.2. Dry Matter Intake Data and Collection and Analysis of Fecal Samples 

The feed offered and the orts were measured daily to calculate the dry matter intake (DMI). Samples of TMR were collected every two days to determine the dry matter (DM) content. TMR samples were placed in the forced-draft oven at 65 °C for 48 h until a constant weight was obtained. Then, the dried samples were milled through a 1 mm screen using a feedstuff mill (KRT-34; KunJie, Beijing, China) and then stored at ambient temperature before further analysis.

On the last day of each experimental period, five cows from each group with similar milk yields were selected for the collection of fecal samples [16]. The same cows were sampled in each group during period I and period II. Fecal samples of each cow were collected at 06:00, 12:00, and 18:00. About 300 g of fecal sample was collected at each time. In the morning, about 10 g fecal samples were sealed in 10 mL conical tubes and immediately frozen in liquid nitrogen and then stored at −80 °C for 16S rRNA sequencing. The remaining fecal samples were mixed and dried for further apparent digestibility analysis.

The DM of TMR and fecal samples was determined by the gravimetric loss of free water after heating at 105 °C for 3 h, as described in the method 950.15 of the Association of Official Analytical Chemists (AOAC) [17]. Neutral detergent fiber (NDF) and acid detergent fiber (ADF) of TMR and fecal samples were analyzed using the ANKOM fiber analyzer (A2000i; American ANKOM, Macedon, NY, USA), as described by Van Soest et al. [18]. Nitrogen was measured according to the method 984.13 (AOAC) [17]. Crude protein (CP) of TMR and fecal samples was calculated by multiplying 6.25 by the content of nitrogen. Ash and ether extract in TMR were measured by the method 924.05 and 920.39 of AOAC, respectively [17]. The acid-insoluble ash (AIA) ratio technique was used to determine the apparent total tract digestibility (ATTD) of dietary nutrients [19]. The AIA in the diets (*Ad*, g/kg) and feces (*Af*, g/kg) was analyzed according to the method described by Keulen and Young [19]. With the concentration of a nutrient in the diet (*Nd*, g/kg) and feces (*Nf*, g/kg), the nutrient ATTD was calculated using the following formula:(1)$$ATTD\;\left(\%\right)=\left[1-\frac{Ad\times Nf}{Af\times Nd}\right]\times 100$$

Fecal bacterial DNA was extracted using HiPure Stool DNA Kits (Magen, Guang zhou, China), according to the manufacturer’s protocols. The V3-V4 region of the 16S rRNA gene was amplified as follows: 95 °C for 2 min, followed by 27 cycles at 98 °C for 10 s, 62 °C for 30 s, and 68 °C for 30 s, and a final extension at 68 °C for 10 min. Former primer 341F (CCTACGGGNGGCWGCAG), reverse primer 806R (GGACTACHVGGGTATCTAAT) [20], and an eight-base sequence-unique barcode were used in this study. Amplicons were extracted from 2% agarose gels, purified using the AxyPrep DNA Gel Extraction Kit (Axygen Biosciences, Union City, CA, USA), and quantified using the ABI StepOnePlus Real-Time PCR System (Life Technologies, Foster City, CA, USA). Purified amplicons were pooled in equimolar mode and sequenced on an Illumina platform to generate paired-end (2 × 250) sequences. To obtain high-quality reads, raw reads were further filtered using FASTP (https://github.com/OpenGene/fastp) according to the following rules: (1) Removing reads containing more than 10% of unknown nucleotides (N), and (2) removing reads containing less than 60% of bases with a quality (Q-value) > 20. Paired-end clean tags were merged as raw tags using FLSAH [21] (version 1.2.11) with a minimum overlap of 10 bp and mismatch error rates of 2%. Noisy sequences of raw tags were filtered by the QIIME [22] (version 1.9.1) pipeline under specific filtering conditions [23]. After removing chimeric tags, high-quality tags were clustered into operational taxonomic units (OTUs) of  ≥97% similarity using the UPARSE [24] pipeline. The tag sequence with the highest abundance was selected as a representative sequence within each cluster. The representative sequences were classified into organisms by a naïve Bayesian model using the RDP classifier [25] (version 2.2) based on the SILVA 132 [26] database (https://www.arb-silva.de/), with confidence threshold values that ranged from 0.8 to 1. The abundance statistics of each taxonomy were visualized using Krona [27] (version 2.6). Alpha diversity indices were calculated in QIIME. Sequence alignment was performed using MUSCLE [28] (version 3.8.31) (http://www.drive5.com/muscle/).

#### 2.2.3. Collection and Analysis of Blood Samples

Blood was collected from the tail vein of cows using 10 mL blood collection heparin-coated tubes (Vacutainer; Becton Dickinson, Franklin Lakes, NanJing) before morning feeding every 14 days. In each group, the five animals that had been used for fecal sampling were used for blood sampling. Blood samples were centrifuged at 3500× *g* at 4 °C for 15 min to obtain plasma and then stored at −20 °C for further analysis.

All of the blood plasma samples were analyzed by a Hitachi 7600 automated biochemistry analyzer (Hitachi Co., Tokyo, Japan). Glucose, insulin, glucagon, β-hydroxybutyric acid (BHBA), triglyceride (TG), glutamic-pyruvic transaminase (GPT), aspartate aminotransferase (AST), alkaline phosphatase (ALP), and total protein (TP) were measured using blood colorimetric commercial kits (DiaSys Diagnostics Systems GmbH, Frankfurt, Germany). Albumin (ALB), globulin (GLB), the total antioxidant capacity (T-AOC), malonaldehyde (MDA), superoxide dismutase (SOD), glutathione peroxidase (GSH-Px), immune globulin A (IgA), immune globulin G (IgG), immune globulin M (IgM), and anti-tumor necrosis factor-α (TNF-α) were measured using bovine ELISA kits (Nanjing Jiancheng Bioengineering Institute, Nanjing, Jiangsu, China).

### 2.3. Statistical Analysis

The DMI, production performance, ATTD, blood parameters, fecal bacterial alpha diversity, and composition data were analyzed using SAS (SAS version 9.4, SAS Institute Inc., Cary, NC, USA). The milk performance and blood parameters were analyzed using the PROC MIXED procedure, with the diet, time, and interaction of diet over time as fixed effects. The DMI, ATTD, and bacterial composition and fecal bacteria alpha diversity were subjected to one-way analysis of variance (ANOVA). Additionally, one-way ANOVA of the bacteria data was based on the normal distribution test. Statistical differences between means were determined by Tukey’s multiple comparison test. A *p*-value of less than 0.01 indicated a highly significant difference. Significance was declared at *p* ≤ 0.05, and tendencies were identified at 0.05 < *p* < 0.10. The fecal bacteria principal coordinates analysis (PCoA) based on the Bray–Curtis distance was conducted and plotted in R software (version 3.6.1).

## 3. Results

### 3.1. DMI, Milk Yield, and Milk Composition

The results of the animal performance and milk composition are shown in Table 2. The DMI, milk yield in period I, 3.5% FCM, and ECM in period II were not affected by treatments or the interaction between treatment and time. The milk fat and protein in the PM1 group were higher than in other groups during period I (*p* < 0.01). The lactose content was not affected by treatments during period I, but the PM2 group had a higher lactose content than other groups during period II (*p* < 0.05). The 3.5% FCM and ECM of the PM1 group in period I were higher than other groups (*p* < 0.01). The SCC of the PM1 group during period I was higher than other groups (*p* < 0.05), while the SCC of the PM1 group during period II was lower than CON (*p* < 0.05) and had no significant difference in comparison to the PM2 group. MUN was higher in the PM-treated groups than the CON group (*p* < 0.05). The ratio of milk yield to DMI in the PM1 group was lower than in other groups during period II (*p* < 0.05), while no significant difference was found during period I.

### 3.2. Apparent Total-Tract Digestibility

The ATTD of the DM, NDF, ADF, and CP of rations in different groups was measured at the end of each period (Appendix A). The results showed that the dietary inclusion of PM silage did not affect the ATTD of TMR.

### 3.3. Blood Parameters

The results of the blood parameters are shown in Table 3. No significant difference was observed in energy metabolism parameters, including GLU, insulin, glucagon, BHBA, and TG, in either period I or II, indicating that PM had no effect on the energy metabolism of lactating dairy cows. GPT was lower, and GLB was higher (*p* < 0.05), in the PM2 group compared with the CON group during period I and AST was lower during period II in treatment groups (*p* < 0.01). TP was increased in the PM2 group during period I (*p* < 0.05). T-AOC and SOD were increased with increased PM silage levels during the experimental period (*p* ≤ 0.01), but not affected by time and the interaction of treatment and time. Furthermore, T-AOC tended to increase as the experimental time changed (*p* = 0.07). MDA was significantly decreased by PM supplementation in period II (*p* < 0.05). IgM was affected by treatment and time in the two periods (*p* < 0.05), with a significant interaction between treatment and time (*p* < 0.05). IgA, IgG, and TNF-α were affected by treatment (*p* < 0.05) during period II, while no interaction was observed between treatment and time.

### 3.4. Fecal Bacteria

A total of 4,369,605 raw reads were obtained from all samples, with an average of 145,654 reads per sample. The total tags number was 4,243,276, with an average tags number per sample of 141,443 and an average length of 441 bp. There was no significant difference among groups in terms of the alpha diversity indices (Chao1, ACE, Simpson, and Shannon index) (Appendix A). No obvious separation was observed among the three groups in the PCoA plot during period I and period II based on the weighted and unweighted unifrac distance (Appendix A).

The most abundant phyla in all samples were *Firmicutes* and *Bacteroidetes* (Appendix A), and the predominant genera were *Ruminococcaceae UCG-005*, *Rikenellaceae RC9 gut group*, and *Ruminococcaceae UCG-010* (Table 4). At the phylum level, no significant difference was observed among the three groups. At the genus level, *Ruminococcaceae UCG-013* and *Tyzzerella-4* were decreased in treatment groups during periods I and II (*p* ≤ 0.05), respectively. 

## 4. Discussion

### 4.1. DMI, Milk Production, and Apparent Digestibility 

The chemical composition of diets can influence ruminants’ DMI, especially the NDF content [29,30]. PM supplementation had no influence on the DMI of cows, which may due to the fact that the diets had a consistent nutritional level. Moreover, different diets similar in chemical composition and digestibility resulted in comparable DMI and milk production [31]. Milk protein production was closely related to dietary CP, rumen fermentable carbohydrate, and rumen microbial protein synthesis [32,33]. Our research indicates that the dietary inclusion of PM silage did not influence the milk protein content of dairy cows, which is a finding consistent with the results of other researchers [16]. The milk composition has also been shown to be affected by cow genes [34], diet [33], management [35], and other factors. We speculate that the observed discrepancy of the PM1 group during period I is the result of the combined effect of these factors. Our findings are in agreement with previous research, which has demonstrated that the milk lactose yield of lactating cows is not influenced when they are fed diets of different forage resources [33]. The diet carbohydrate composition and content have been shown to be the key factors that affect the lactose yield via glucose metabolism [36]. The findings regarding the lactose yield are attributed to the fact that, in our study, all diets had similar concentrations of NDF and ADF. Lower SCC in milk indicates a better health condition of dairy cows and better milk quality [37]. The result that PM could lower the milk SCC is consistent with prior research [16], which could be the effect of flavonoids in the PM [4], since flavonoids can reduce the milk SCC [38] by enhancing the immunity of dairy cows [39].

ATTD can be affected by feedstuff resources [32], the dietary chemical composition [40], and the physiological stage of cows [41]. In this study, the different feedstuff resources were the key factor that affected the digestibility. The NDF content had a negative correlation with digestibility [42]. DMI was also an important factor influencing the DM digestibility. The decline in DM digestibility could mostly be accounted for by simultaneous increases in the level of feed intake [43]. The nutritional level of different diets was similar and the equal DMI explains the lack of difference in ATTD within different groups.

### 4.2. Blood Parameters

All blood parameters were within a normal range and the treatment groups had a similar trend to prior studies, which also presented dairy cows with different levels of PM silage [8,16]. Blood glucose, insulin, and BHBA are the key indicators of the energy metabolism of cows [8], and they can be affected by the cow’s physiological stage and diet energy content [8,44]. In the current study, the diets in different groups had the same energy level and the experimental cows were in a similar lactation stage. All of these contributed to the accordance of the energy metabolism status. The results of the antioxidant parameters in our study are consistent with Si et al. [16]. The antioxidant substance level in shrubs and trees is higher than grass [9]. Ruminants’ intake of antioxidant-rich herbs as feed supplements can increase the blood T-AOC [9]. Differences between diets in terms of the antioxidant parameters can be attributed to alkaloids, flavonoids, anthocyanins, and some polyphenols, which have been reported to occur at high levels in PM silage [4]. Flavonoids can enhance the anti-inflammatory effect of ruminants [10,39]. Additionally, anthocyanins can improve the antioxidant and immune capacity of cows [45]. The underlying mechanism is that the bioactive substances in PM silage, such as flavonoids, act as a reducing agent and a hydrogen donor to neutralize oxygen free radicals and remove hydrogen peroxide and superoxide ions [6,10]. MDA is the final product of lipid peroxidation [46]. Jin et al. found that alfalfa flavonoids could lower the blood MDA content of dairy cows [47], which is consistent with our results. The low concentraiton of MDA in PM has also been associated with less peroxidation activity in dairy cow’s body metabolism [46]. Our findings regarding the liver function parameters were also consistent with previous reports [16], which found that PM silage could improve the liver function of dairy cows. The impairment of liver function can lead to an increase of the blood concentrations of GPT and AST [48]. ALB is also related to impairment of the liver function due to inflammatory conditions [49]. A greater GLB concentration also indicates a better immune responsiveness of dairy cows [50]. Hence, evidence indicates that PM silage can improve the antioxidant capacity by increasing the activities of antioxidant enzymes, thereby protecting tissues and cells against damage mediated by free radicals.

### 4.3. Fecal Bacteria

Previous studies have shown that the fecal bacterial community is related to the health and antibiotic activity of cattle [12]. Here, we further evaluated the effects of PM silage on the fecal bacterial community in lactating dairy cows. In accordance with previous studies [51,52], our study also found that *Firmicutes* and *Bacteroidetes* are the two dominant phyla in dairy cows’ feces. The intestinal bacteria are related to the host health and production performance [11,12]. In the current study, there were no differences between dietary treatments in the fecal bacterial structure. This finding was expected because all diets had a similar chemical composition and the dietary chemical composition has been identified as the main driver of the gastrointestinal microbiome [13]. One study using mulberry leaf in the diet of finishing steer found that the fecal domain genus bacteria were different from our results [53]. The fecal bacteria composition is also related to the feeding operation [54] and host genetics [55]. The different breed and feeding management can be attributed to the inconsistency of the domain genus. The *Ruminococcaceae* family is related to cellulose and hemicellulose degradation [56]. Diets with PM silage changed the carbohydrate resource and the fiber decomposition process may contribute to the difference between the *Ruminococcaceae UCG-013* genus and other *Ruminococcaceae* genera [56]. Previous studies have indicated that the large intestine of cattle contains a small quantity of fiber and starch [57], and that the large intestine is an important organ for digesting undigested feed, suggesting that PM silage supplementation can have a positive influence on fiber degradation in the upper gut and consequently decrease the cellulolytic bacteria of the hindgut. *Tyzzerella-4* is a potentially pathogenic bacteria and animals fed with probiotics can exhibit a reduced *Tyzzerella-4* content in the intestine [58]. The decreased *Tyzzerella-4* genus in the PM treatment groups may relate to the fact that PM silage is rich in bioactive substances [3] and can further enhance the intestinal health [10].

## 5. Conclusions

Diets with different levels of PM silage did not influence the TMR ATTD and daily milk yield of lactating dairy cows in the second lactation period. PM silage can strengthen the antioxidant ability and immunity of cows, which may be attributed to the bioactive substance in PM. Diets with PM silage had a limited influence on dairy cows’ fecal bacteria. With the shortage of feedstuff and the prohibition of antibiotic use in animals, PM can be a functional feedstuff for ruminants. The specific mechanism of bioactive substances in PM to improve the health of dairy cows requires further study.

## Figures and Tables

**Table 1 animals-10-01152-t001:** The ingredient and nutrient levels of experimental diets.

Item ^1^	Treatment					
	Period I			Period II		
Groups	CON	PM1	PM2	CON	PM1	PM2
Ingredients, % DM						
Oat	0	0	0	4.1	4.0	6.1
Alfalfa hay	6.0	4.0	4.0	6.2	4.0	0
Alfalfa silage	8.6	5.7	2.8	7.4	0	0
Paper mulberry silage	0	4.5	9.0	0	13.5	18
Whole corn plant silage	38.5	38.5	38.0	34.7	30.0	27.3
Steam-flaked corn	13.6	13.6	13.6	12.0	12.6	12.6
Whole cottonseed	2.1	2.1	2.1	2.1	2.1	2.1
Soybean meal	13.2	13.2	13.2	15.3	15.3	15.3
Corn	9.3	9.3	9.3	9.4	9.4	9.4
Bran	1.5	1.5	1.5	1.5	1.5	1.5
Extruded soybean meal	1.1	1.1	1.1	1.1	1.1	1.1
DDGS	2.1	2.1	2.1	2.2	2.2	2.2
Sodium bicarbonate	1.1	1.1	1.1	1.1	1.1	1.1
Magnesium oxide	0.3	0.3	0.3	0.3	0.3	0.3
Yeast	0.3	0.3	0.3	0.3	0.3	0.3
Montmorillonite	0.1	0.1	0.1	0.1	0.1	0.1
Mineral vitamin premix ^2^	2.1	2.1	2.1	2.1	2.1	2.1
Chemical levels, % DM						
CP	17.1	17.3	17.2	17.2	17.2	17.2
Ether extract	4.5	4.7	4.7	4.6	4.5	4.6
Ash	5.9	5.7	6.1	5.8	6.0	5.9
NDF	29.8	30.0	30.1	30.4	30.7	30.3
ADF	20.1	20.3	20.3	20.4	20.5	20.6
NEL ^3^, (Mcal/kg)	1.7	1.7	1.7	1.7	1.7	1.7

^1^ DM: dry matter; DDGS: dry distilled grain soluble; CP: crude protein; NDF: neutral detergent fiber; ADF: acid detergent fiber; NEL: net energy for lactation. ^2^ Premix provided the following per kg: 6000 mg Mn; 4800 mg Fe; 9000 mg Zn; 2600 mg Cu; 120 mg I_2_; 80 mg Se; 70 mg Co; 130,000 IU Vitamin A; 336,000 IU Vitamin D; and 465 IU Vitamin E. ^3^ NEL was a calculated value, while the others were measured values.

**Table 2 animals-10-01152-t002:** Effects of paper mulberry silage on the dry matter intake (DMI) and milk yield of dairy cows.

Item ^1^	Period	Treatment ^2^			SEM	*p*-Values		
		CON	PM1	PM2		Treatment	Time	Treatment × Time
DMI, kg/day	І	23.2	23.0	22.9	0.3	0.83	-	-
	ІІ	22.9	23.0	22.8	0.5	0.68	-	-
Milk yield, kg/day	І	36.1	36.0	35.9	4.4	0.87	0.97	1.00
	ІІ	33.9 ^a^	32.8 ^b^	33.8 ^a^	5.4	0.04	0.99	1.00
3.5% FCM, kg/day	І	38.9 ^b^	40.7 ^a^	38.3 ^b^	6.1	<0.01	0.72	1.00
	ІІ	37.3	36.3	37.0	6.2	0.17	0.41	1.00
ECM, kg/day	І	39.2 ^b^	41.1 ^a^	38.7 ^b^	5.8	<0.01	0.98	1.00
	ІІ	37.7	36.6	37.4	6.4	0.10	0.17	1.00
Milk composition, %								
Protein	І	3.6 ^b^	3.7 ^a^	3.6 ^b^	0.3	<0.01	<0.01	1.00
	ІІ	3.7	3.6	3.6	0.4	0.54	<0.01	0.99
Fat	І	4.0 ^b^	4.3 ^a^	3.9 ^b^	0.1	<0.01	0.92	0.99
	ІІ	4.1	4.2	4.1	0.5	0.23	0.11	0.38
Lactose	І	5.1	5.1	5.1	0.2	0.11	<0.01	0.34
	ІІ	4.9 ^b^	5.0 ^b^	5.1 ^a^	0.6	<0.01	0.02	0.98
MUN	І	10.5 ^b^	11.0 ^a^	11.2 ^a^	1.3	<0.01	<0.01	0.90
	ІІ	15.0 ^b^	17.3 ^a^	17.1 ^a^	1.3	<0.01	1.00	<0.01
SCC	І	16.6 ^b^	26.3 ^a^	14.0 ^b^	27.5	<0.01	1.00	0.90
	ІІ	22.2 ^a^	15.0 ^b^	18.2 ^b^	22.4	<0.01	0.60	1.00
Efficiency, kg/kg								
Milk yield/DMI	І	1.6	1.6	1.6	0.2	0.58	0.99	1.00
	ІІ	1.5 ^a^	1.4 ^b^	1.5 ^a^	0.2	0.01	0.99	1.00
3.5% FCM/DMI	І	1.7 ^b^	1.8 ^a^	1.7 ^b^	0.1	<0.01	0.92	1.00
	ІІ	1.6	1.6	1.6	0.1	0.23	0.11	0.38

^a,b^ Least squares means within a row with different superscripts differ significantly (*p* ≤ 0.05). ^1^ DMI: dry matter intake; FCM: fat-corrected milk; ECM: energy-corrected milk; MUN: milk urea nitrogen; and SCC: somatic cell count. 3.5% FCM (kg/day) = 0.432 × milk yield + 16.23 × fat yield; ECM (kg/day) = 12.82 × fat yield + 7.13 × protein yield + 0.323 × milk yield. ^2^ The CON group had non-PM silage diets during period I and period II; the PM1 group had 4.5 and 13.5% PM silage diets during period I and period II, respectively; and the PM2 group had 9 and 18% PM silage diets during period I and period II, respectively.

**Table 3 animals-10-01152-t003:** Effects of paper mulberry silage on the blood parameters of dairy cows.

Item ^1^	Period	Treatment ^2^			SEM	*p*-Values		
		CON	PM1	PM2		Treatment	Time	Treatment × Time
Energy metabolism								
GLU, mmol/L	І	3.3	3.3	3.4	0.4	0.67	0.35	0.90
	ІІ	2.8	2.9	2.8	0.3	0.74	0.94	0.03
Insulin, mIU/mL	І	17.5	20.4	20.1	5.7	0.45	0.34	0.35
	ІІ	17.6	18.4	20.4	4.7	0.50	0.78	0.79
Glucagon, pg/mL	І	151.2	151.3	153.0	26.6	0.96	0.49	0.40
	ІІ	182.4	174.0	172.4	23.1	0.63	0.30	0.88
BHBA, mmol/L	І	0.4	0.6	0.5	0.2	0.23	0.24	0.54
	ІІ	0.5	0.7	0.7	0.2	0.11	0.68	0.08
TG, mmol/L	І	0.3	0.3	0.3	<0.1	0.73	0.13	0.14
	ІІ	0.3	0.3	0.3	<0.1	0.97	0.23	0.25
Liver function								
GPT, U/L	І	27.8 ^a^	25.9 ^a^	20.0 ^b^	3.4	<0.01	0.18	<0.01
	ІІ	26.3	25.9	27.3	3.4	0.71	0.56	0.10
AST, U/L	І	73.8	80.1	70.3	13.4	0.32	0.85	0.99
	ІІ	91.0 ^a^	81.8 ^a,b^	70.4 ^b^	12.5	0.01	0.98	1.00
ALP, U/L	І	47.7	49.1	48.7	6.9	0.95	0.43	0.15
	ІІ	53.1	59.4	55.8	10.2	0.55	0.98	0.82
TP, g/L	І	65.3 ^b^	62.6 ^b^	70.6 ^a^	4.2	<0.01	0.96	0.39
	ІІ	67.1	67.3	73.4	6.7	0.08	0.83	0.85
ALB, g/L	І	33.8	33.2	33.8	2.3	0.80	0.76	0.31
	ІІ	33.9	35.8	34.7	1.8	0.08	0.84	0.68
GLB, g/L	І	30.4 ^b^	31.0 ^b^	35.3 ^a^	4.5	0.05	0.37	0.32
	ІІ	33.2	31.5	34.4	5.3	0.54	0.98	0.91
Oxidative stress								
T-AOC, U/mL	І	24.9 ^b^	29.2 ^a^	29.8 ^a^	2.0	<0.01	0.15	0.93
	ІІ	22.4 ^b^	25.3 ^a^	25.9 ^a^	2.3	0.01	0.07	0.19
MDA, nmol/mL	І	5.1	5.6	4.7	2.3	0.75	0.38	0.24
	ІІ	5.1 ^a^	4.4 ^a,b^	4.0 ^b^	0.6	0.02	0.21	0.92
SOD, U/mL	І	50.5 ^b^	50.8 ^b^	53.8 ^a^	2.1	<0.01	0.15	0.85
	ІІ	54.0 ^b^	54.7 ^b^	57.0 ^a^	2.1	<0.01	0.96	0.67
GSH-Px, U/mL	І	317.1	328.5	316.8	52.1	0.86	0.01	0.63
	ІІ	329.3	313.3	332.3	34.2	0.71	0.19	0.28
Immunity parameters								
IgA, μg/mL	І	240.4	260.6	260.8	28.0	0.43	0.59	0.02
	ІІ	241.8 ^b^	227.3 ^b^	296.5 ^a^	48.1	0.03	0.90	0.41
IgG, mg/mL	І	8.2	7.5	9.0	1.0	0.06	0.53	0.01
	ІІ	8.1 ^b^	7.8 ^b^	10.1 ^a^	1.4	0.01	0.41	0.17
IgM, mg/mL	І	2.3 ^b^	2.4 ^b^	2.9 ^a^	0.3	0.01	0.01	0.02
	ІІ	2.5 ^b^	2.8 ^b^	3.3 ^a^	0.5	0.03	<0.01	0.75
TNF-α, ng/L	І	531.8	508.4	515.9	52.4	0.63	0.33	0.38
	ІІ	545.4 ^a,b^	515.2 ^b^	593.4 ^a^	47.9	0.04	0.09	0.11

^a,b^ Least squares means within a row with different superscripts differ significantly (*p* ≤ 0.05). ^1^ GLU: glucose; BHBA: β-hydroxybutyric acid; TG: triglyceride; GPT: glutamic-pyruvic transaminase; AST: aspartate aminotransferase; ALP: alkaline phosphatase; TP: total protein; ALB: albumin; GLB: globulin; T-AOC: total antioxidant capacity; MDA: malonaldehyde; SOD: superoxide dismutase; GSH-Px: glutathione peroxidase; IgA: immune globulin A; IgG: immune globulin G; IgM: immune globulin M; and TNF-α: anti-tumor necrosis factor -α. ^2^ The CON group had non-PM silage diets during period I and period II; the PM1 group had 4.5 and 13.5% PM silage diets during period I and period II, respectively; and the PM2 group had 9 and 18% PM silage diets during period I and period II, respectively.

**Table 4 animals-10-01152-t004:** Effects of paper mulberry silage on the fecal bacteria composition at the genus level (top fifteen).

Items	Period	Treatment ^1^			SEM	*p*-Values
		CON	PM1	PM2		
*Ruminococcaceae UCG-005*	I	14.5	14.8	17.5	0.8	0.33
	II	15.6	18.0	17.1	0.6	0.40
*Rikenellaceae RC9 gut group*	I	10.0	11.5	14.2	1.0	0.08
	II	10.2	10.7	11.5	0.3	0.45
*Ruminococcaceae UCG-010*	I	9.2	9.4	9.6	0.1	0.89
	II	9.7	9.5	9.5	0.1	0.91
*Coprostanoligenes group*	I	6.2	6.1	4.8	0.4	0.40
	II	5.7	4.8	4.6	0.3	0.43
*Prevotellaceae UCG-003*	I	4.9	5.0	4.5	0.1	0.51
	II	3.9	3.9	4.2	0.1	0.73
*Alistipes*	I	3.6	4.1	4.2	0.1	0.22
	II	4.0	3.4	4.0	0.2	0.14
*Bacteroides*	I	3.7	3.4	3.1	0.3	0.76
	II	2.7	2.8	3.0	0.1	0.91
*Ruminococcaceae UCG-013*	I	3.6 ^a^	3.3 ^b^	2.2 ^b^	0.3	0.05
	II	3.0	3.1	2.3	0.2	0.33
*Christensenellaceae R-7 group*	I	2.0	1.9	1.7	0.1	0.33
	II	1.8	1.6	1.6	0.1	0.26
*Treponema-2*	I	1.6	1.7	1.4	0.1	0.85
	II	2.1	2.2	1.6	0.2	0.61
*Ruminococcaceae UCG-014*	I	2.1	1.8	1.0	0.3	0.10
	II	2.2	1.4	1.2	0.3	0.09
*Tyzzerella-4*	I	1.1	0.9	0.9	0.1	0.54
	II	1.2 ^a^	0.8 ^b^	1.0 ^b^	0.1	0.05
*Ruminococcaceae UCG-009*	I	0.9	1.0	0.6	0.1	0.09
	II	0.9	1.0	0.6	0.1	0.06
*Erysipelotrichaceae UCG-004*	I	0.3	0.4	0.6	0.1	0.11
	II	0.5	0.3	0.6	0.1	0.06
*Alloprevotella*	I	0.4	0.6	0.4	0.1	0.45
	II	0.4	0.3	0.3	<0.1	0.06

^a,b^ Least squares means within a row with different superscripts differ significantly (*p* ≤ 0.05). The fecal bacteria composition was expressed as a percentage of the total number of sequences. ^1^ The CON group had non-PM silage diets during period I and period II; the PM1 group had 4.5 and 13.5% PM silage diets during period I and period II, respectively; and the PM2 group had 9 and 18% PM silage diets during period I and period II, respectively.

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
