# Peer review of "Effects of Paper Mulberry Silage on the Milk Production, Apparent Digestibility, Antioxidant Capacity, and Fecal Bacteria Composition in Holstein Dairy Cows"

_animals, 2020, doi:10.3390/ani10071152_

Round 1

Reviewer 1 Report

This is an interesting manuscript regarding a topic of local interest in the paper mulberry growing areas of China. The research appears to have been appropriately conducted and for the most part, the conclusions are justified. There are several issues with the paper that should be addressed:

  1. English language. This reviewer is impressed that the authors are able to write scientific language to this standard. Nevertheless, the manuscript still needs considerable improvement in English. In most cases I am able to understand the ideas put forth by the authors, even though there is considerable incorrect word usage. However, in some cases, there are sentences that I just do not understand what the authors are trying to convey. It is suggested that the authors involve a native English speaker in the writing of this manuscript.
  2. This manuscript is about “paper mulberry”. It is stated (Line 15 in the simple summary) that paper mulberry is a woody roughage. Yet, in this experiment, paper mulberry replaced alfalfa hay and alfalfa silage with no effect on diet Dry matter intake. Furthermore, the diet chemical compositions were all similar with respect to NDF %. Thus, it seems that in this case the paper mulberry was not very woody at all. Most western readers will not be familiar with paper mulberry. It is suggested that the authors include an additional table listing the chemical composition of paper mulberry.
  3. Excessive precision. In most of the tables, mean values for many items are listed with excessive precision. For example, in Table 2 CON, DMI is listed as 23.17, but 23.2 would have been appropriate. It is suggested that in most cases, 3 figures (numbers) are sufficient. For example, in Table 4, period I, Glucagon is listed as 151.21, when 151 would have been sufficient and the SEM of 26.6.

Reviewer 2 Report

Abstract, line 32: delete (p > 0.05). Check allover the text if you wrote in other sections p > 0.05; it is not useful to the readers, can be p = 0.06 or p = 0.96

Materials and methods, line 77: write "two times/d" not daily.

Author Response

Dear professor,

First of all, thank you for your review, and give me good advice to make my paper more perfect. I have modified them one by one according to your comments and then answer your questions here.

  1. Question:

Abstract, line 32: delete (p > 0.05). Check allover the text if you wrote in other sections p > 0.05; it is not useful to the readers, can be p = 0.06 or p = 0.96

    Answer:

I have deleted the "p > 0.05" in the abstract and the parts of the result. I have read through the manuscript and deleted other descriptions like this in the whole paper.

  1. Question:

Materials and methods, line 77: write "two times/d," not daily.

     Answer:

I have change the "two times daily" into "two times/d."

That's all. Thanks for the excellent advice again. At the same time, I have revised my paper according to other reviewers.

Best wishes

Yangyi Hao

Reviewer 3 Report

The article entitled " Effects of paper mulberry silage on milk production, apparent digestibility, antioxidant capacity and fecal bacteria composition in Holstein dairy cows” explains the potential that mulberry silage may have in increasing antioxidant capacity and immunity of dairy cows. And it concludes that there is no impact (positive or negative) on milk production, digestibility and fecal bacteria composition.  It seems that mulberry silage can be interesting to use with a positive impact on dairy cows’ health and very interesting for the farming system. However, this reviewer thinks that the article presented needs to improve and expand upon some critical aspects which will be detailed (attached file):

MAIN CONCERNS:

  • Introduction: The reader understands that it is a very common tree in China and its cultivation is focused on silage for cows. Is it like that? Its cultivation is only focused on silage? Or silage is a secondary use? What part of the tree is collected for silage? Leaves, fruits, etc...It is important to describe it as a byproduct or not...I assumed it is leaves only (line 60), but please, be more consistent everywhere in the paper (Title and abstract included).
  • Material and Methods section needs much more detailed info, above all in study design, this is critical. Above all, about the decision of grouping and, after 28 days, increase PM % in the experimental groups (cumulative effect possible?). Also has to be well clarified why only the top 5 cows of each group were blood and fecal sampled. I concluded these cows were also different for period 1 and 2, so groups explained in the paper (15 cows per group?) were only for milk yield data?. Please, rewrite and better explain the M&Ms Section and check the detailed list of this review.
  • Discussion is confusing because some parts should be in M&Ms. Please, rewrite and check the detailed list.
  • It is necessary to justify very well why only the 5 most productive animals of each group have been analyzed in the laboratory. Obviating the rest of the cows, which at least at the beginning of the study had very similar productions, could have provided more information and much more conclusive? At the very least, this should be very well explained in M & Ms and explained in Discussion.

Author Response

Dear reviewer,

First of all, thank you for your time to review our paper. I appreciate your suggestions very much. I have modified your mentioned issues one by one in the manuscript according to your comments, and please find below my answers to your questions.

Question

   Introduction: The reader understands that it is a very common tree in China, and its cultivation is focused on silage for cows. Is it like that? Its cultivation is only focused on silage? Or silage is a secondary use? What part of the tree is collected for silage? Leaves, fruits, etc...It is important to describe it as a byproduct or not...I assumed it is leaves only (line 60), but please, be more consistent everywhere in the paper (Title and abstract included).

Answer:

Thank you for your suggestions. Indeed, western researchers would not be familiar with paper mulberry. Therefore, I have given some more general introduction about paper mulberry, as well as some information if it is used as roughage. At the same time, I have given the chemical composition data of paper mulberry silage in the supplement files.

In this study, we used the whole plant paper mulberry silage for the experiment. Before our work, most of the researchers used the leaves to feed animals. Therefore, I referred to their works in my introduction part. Originally, paper mulberry were planted in China to prevent soil erosion, because it can grow very fast. Recently, people started to realize that PM is rich in protein, so it has been started to use PM as feed raw material. For pig and chicks, they use the leaves of paper mulberry. I hope this explanation could clear your confusion somehow. You can also find the modified information in the listed parts in the manuscript:

Paper mulberry (Broussonetia papyrifera; PM) is a dioecious tree native to mainland South-Asia and East Asia [1]. Twenty years ago, the PM was planted to prevent soil erosion in China. In recent years, it has been recognized that PM is fast growth and has high protein content, which can be used as an excellent roughage resource for ruminant animals. So far, about 300 000 hectares of PM are currently under cultivation in China, and most of them were planted as forage crops and processed into silage. Usually, the leaves of PM were fed to the pigs and chicks, and the whole plant can be silage and fed to ruminants. The PM can be harvest three to five times per year in China. Furthermore, the process techniques were mature for PM silage.

We also added the chemical composition of paper mulberry silage we used in this study in the supplement material files and. The details are also listed below.

Table S1 The chemical composition of paper mulberry silage

Item1

DM, as-fed basis

NDF

ADF

CP

Content,%

20.4

43.1

33.5

16.9

  • DM: dry matter, NDF: neutral detergent fiber, ADF: acid detergent fiber, CP: crude protein.

Question

Material and Methods section needs much more detailed info, above all in study design, this is critical. Above all, about the decision of grouping and, after 28 days, increase PM % in the experimental groups (cumulative effect possible?). Also has to be well clarified why only the top 5 cows of each group were blood and fecal sampled. I concluded these cows were also different for period 1 and 2, so groups explained in the paper (15 cows per group?) were only for milk yield data?. Please, rewrite and better explain the M&Ms Section and check the detailed list of this review.

Answer:

The grouping criterion. We selected the 2±0.4 lactation dairy cows. Compare to the primiparous cows, the milk yield and body metabolism of the multiparous cows are more stable. So we think these cows can better represent the lactating cows. Therefore, the average lactation parity of dairy cows is about 2.7 in China. For high yield farm, the average lactation parity of dairy is lower. The 2nd parity is the most common stage of lactating dairy cows in China. There are not enough 3rd lactation parity cows. The first lactation cows, sometimes we called them as heifer, had to grow it’s body size meanwhile milking. We think the 2ed lactation cows can represent the adult milking cow. So we chose 2ed lactation cows to sample.

We didn't select the top five cows. We chose the cows with similar milk yield within the three groups for fecal and blood samples collection. That's because we thought five cows could represent their group, and we had sampled every fourteen days. This means each experiment period had ten cows be sampled. At the same time, honestly, the cost of blood samples and fecal bacteria was prohibitive, and we can't afford it. There also many dairy cow feeding experiments that sampled parts of the whole experiment cows. Like the published article: “Effect of Broussonetia papyrifera L. (paper mulberry) silage on dry matter intake, milk composition, antioxidant capacity and milk fatty acid profile in dairy cows.”

After the first 28 d experiment, we found there were no effection on the production performance of dairy cows. Therefore we added 9.0% PM silage in the treatment groups and gave them seven days pre-feeding period. I have explained this point in the paper as follows:

To evaluate the effects of feeding different levels (4.5, 9.0, 13.5, and 18.0%) of PM silage on dairy cows, the experiment was divided into two periods. Length of the adaption period was 7 days, and the formal experimental period was 21 days for both periods.

Question

   Discussion is confusing because some parts should be in M&Ms. Please, rewrite and check the detailed list.

Answer: Thanks for your advice on my discussion parts. I have rewritten this part and changed the wrong format words in the article. I also deleted some sentences like materials and methods.

That's all. Thanks for the excellent advice again. At the same time, I also revised my paper according to other reviewers.

Round 2

Reviewer 1 Report

The manuscript is much improved. A small number of improvements in English are still required, and suggested edits are listed below:

Line 15-18:  ……. PM) is a roughage rich in bioactive substances such as phenolics and flavonoids which are beneficial for animal health. This study evaluated the apparent digestibility ….

L 27-28: Forty-five lactating Holstein dairy cows with ……

L 30: Then, treatment groups were fed diets containing ….

L 31-33: Paper mulberry silage increased milk urea nitrogen and decreased the somatic cell count (p < 0.05), but did not affect dry matter intake …..

L 36-38: In conclusion, PM silage enhanced the antioxidant capacity and immunity of dairy cows, but did not influence milk yield, dry matter digestibility and fecal bacterial composition.

L 44-45:  …. has high protein content, which makes it useful as a roughage resource for ruminant animals.

L46:  ….. were planted as forage crops intended to be processed into silage.

L 47-49:  …. Usually the leaves of PM are fed to pigs and chickens, while the whole plant is processed into silage which is fed to ruminants. The PM can be harvested three to …..

L 51: …. On the non-structural carbohydrate and ….

L54: It is rich inpolyphenols such as flavonols, ………

L 74: supplemented with PM silage could ……

L75: Therefore, the objectoive of our study was to evaluate the apparent ……

L80-81: The feeding experiment was conducted at Aomei Dairy Farm in Xinxiang City, Henan province, China, using dairy cows with an average 305-day milk yield of 900 kg per cow.

L92-93: The chemical composition of PM silage is presented in Table S1 while the ingredients and nutritional levels of the diets are presented in Table 1. All cows had access to feed and water ad libitum.

L96: Table 1 The ingredient and nutrient levels of experimental diets.

L103: Collection and analysis of milk samples

L108: Dry matter intake data and collection and analysis of fecal samples

L113:  and then stored at ambient temperature before further analysis.

L115: selected for collection of fecal samples [16]. The same cows were sampled in each group during period I and period II.

L119: The remaining fecal samples were …….

L 156: 2.2.3 Collection and analysis of blood samples

L 158-159: In each group, the five animals that had been used for fecal sampling were used for blood sampling.

L 195: Table 2 Effects of paper mulberry silage on DMI and milk yield of dairy cows

L 205: The ATTD of DM, NDF, ADF, and CP of rations in different groups were measured at the end of each period (Table S2).

L206-207: Dietary inclusion of PM silage did not affect the ATTD of TMR.

L211: ….. indicating that PM had no effect on the …..

L255-256:  ….Our research indicates that dietary inclusion of PM silage does not influence milk protein content of dairy cows, a finding consistent with the findings of other researchers [16].

L257-258: Milk composition has also been shown to be affected by genetics [34], diet [33], management [35] and other factors.

L258: We speculate that the observed discrepancy of PM1 group during period I is the result of the

combined effect of these factors.

L259-260  Our findings are in agreement with previous research that has demonstrated that the milk lactose yield of lactating cows is not influenced when they are fed diets of different forage sources [33].          [Note, previous research cannot be in agreement with your research because your research had not occurred when there research was conducted. But your research findings can be in agreement with previous research]

261: The diet carbohydrate composition and content have been shown to be the key factor that affect the lactose yield via glucose metabolism [36].

262-263: The findings regarding lactose yield are attributed the fact that in our study, all diets had similar concentrations of NDF and ADF..

L 274: Blood Parameters

L278: In the current study,

L 283-284: Differences between diets in the antioxidant parameters can be attributed to alkaloids, flavonoids, anthocyanins, and some polyphenols, which have been reported to occur at high levels in PM silage [4].

L 290: which is consistent with our results. The low concentration of MDA in PM has also been associated with less peroxidation activity in dairy cow’s body metabolism [46].

L 291-293: Our findings regarding liver function parameters were also consistent with previous reports [16] that found PMsilage could improve the liver function of dairy cows.

L293: The impairment of liver function can lead to an increase in blood concentrations of GPT and AST [48].

295: The greater GLB concentration is also indicative of better immune responsiveness of dairy cows.

L304-307: In the current study, there were no differences between dietary treatments in the fecal bacterial structure. This finding was expected because all diets had similar chemical composition and dietary chemical composition has been identified as the main driver of the gastrointestinal microbiome[13].

Author Response

Dear reviewer,

    First of all, thank you again for your time in correcting our paper. I appreciate your patience in giving me specific writing advice. I have modified your mentioned issues one by one in the manuscript according to your comments.

    Special thanks for your note about “our research findings can be in agreement with previous research, not their research is in agreement with us.” It is my lucky encounter with such a good reviewer, not only tell me where there is a problem, but also give me why.

   Thank you again.

                                                Best wishes

                                                Yangyi Hao

Reviewer 3 Report

I am pleased to say that the quality of this manuscript has greatly increased in all aspects. It has also improved the quality of the figures provided in this new version. However, I think that english must be reviewed, still.

 Despite the substantial increase in manuscript quality, I will go into some minor comments:

Line 80: Please add info about fat, protein and Avg Somatic cell count.

Line 84: Daily milk yield

Line 92: how many times did you analyze the ratio chemical composition in every period? Please explain. If it was more than once, then explain that data are Average chemical composition...

Table 1 and for all the manuscript: Please, choose/decide to use 1 or 2 decimals and be consistent.

Table 1:

Add “Group” in the line between “item” and “Ingredients”

Add DM definition in the legend.

Line 95: How did you collect the milk yield? Directly in the milking parlor? software?

Line 276: prior studies? you are referencing to TWO studies...

Line 277: ... are key indicators...

Line 279: Lactation stage

Line 281: consistent

Line 322: different

Line 322: daily milk yield

Author Response

Dear reviewer,

First of all, thank you again for your time in correcting our paper. I have modified your mentioned issues one by one in the manuscript according to your comments. The following is some detailed reply to your question.

Question: Line 80: Please add info about fat, protein and Avg Somatic cell count.

Answer: I have given more information about the experimental farm. It was as follows: The feeding experiment was conducted at Aomei Dairy Farm in Xinxiang City, Henan province, China, using dairy cows with the average 305-day milk yield per cow, milk fat content, milk protein content, and the somatic cell count (SCC) of 9 000 kg, 3.9 %, 3.4 %, and 20 000/mL respectively.

In 2019, the average milk yield of dairy cows was about 7 000 kg per cow per year in China. Our farm is a little higher than the average.

Question: how many times did you analyze the ratio chemical composition in every period? Please explain. If it was more than once, then explain that data are Average chemical composition...

Answer: I have analyzed the ratio chemical composition four times. In other words, I analyzed the ratio once every 14 days. So I write in the paper as follows: The chemical composition of PM silage and other fodders were analyzed once every 14 days. The average chemical composition of PM silage is presented in Table S1, while the ingredients and nutrient levels of the diets are presented in Table 1.

Question: Table 1 and for all the manuscript: Please, choose/decide to use 1 or 2 decimals and be consistent.

Answer: Thanks for your kind advice. To be honest, all the data in the manuscript is using two decimal at first. But the reviewer 2 advice me to use three figures during the first time review. His advice is as follows:

“In most of the tables, mean values for many items are listed with excessive precision. For example, in Table 2 CON, DMI is listed as 23.17, but 23.2 would have been appropriate. It is suggested that in most cases, 3 figures (numbers) are sufficient. For example, in Table 4, period I, Glucagon is listed as 151.21, when 151 would have been sufficient and the SEM of 26.6.”

So I have changed the data into 3 figures, for example: 151.21 into 151, 23.17 into 23.2 in the previous modification. When I received your advice, I read about 10 papers which have published in animals. I found some data in these papers with 1 decimal, some data with 2 decimals, even some data had no decimal. In the same article, the number of data decimals is also different. Finally, I decided to change all the data with 1 decimal. I also in agreement with you; all the data with 1 decimal make the paper more consistent.

Question: Line 95: How did you collect the milk yield? Directly in the milking parlor? software?

Answer: In fact, I recorded the milk yield manually. So I added the word: “manually” in my paper. The sentence is as follows: Milk yield data was recorded manually every day.

Thank you again.

Best wishes

Yangyi Hao